# Understanding the Healthcare Needs of Immigrant Children Currently and Previously in Government Custody: A Narrative

**Jaime La Charite** [1,2,3], **Elizabeth W. Tucker** [4], **Julia Rosenberg** [5], **Janine Young** [6], **Nikita Gupta** [1] **and Katherine Hoops** [7,8,*]

1   Department of Pediatrics, Johns Hopkins University School of Medicine, Baltimore, MD 21205, USA
2   Division of General Internal Medicine and Health Services Research, Department of Internal Medicine, University of California, Los Angeles, CA 90024, USA
3   Department of Internal Medicine and Pediatrics, Cedars-Sinai Medical Center, Los Angeles, CA 90048, USA
4   Department of Anesthesiology and Critical Care, Johns Hopkins University School of Medicine, Baltimore, MD 21205, USA
5   Department of Pediatrics, Yale University School of Medicine, New Haven, CT 06510, USA
6   Department of Pediatrics, University of California San Diego School of Medicine, San Diego, CA 92093, USA
7   Department of Anesthesiology and Critical Care Medicine, Johns Hopkins University School of Medicine, Baltimore, MD 21205, USA
8   Department of Health Policy and Management, Johns Hopkins Bloomberg School of Public Health, Baltimore, MD 21205, USA
*   Correspondence: khoops1@jh.edu

**Abstract:** Little is known of pediatric clinicians' experiences with and approaches to taking care of immigrant children who have been in US custody. The objectives of this article are to (1) recognize the challenges facing pediatric clinicians in caring for immigrant children previously in custody, and (2) propose ways that healthcare and legal professionals can collaborate to optimize the wellbeing of formerly detained immigrant children. We identify themes by assessing answers to multiple choice and short responses from a national survey. These findings can help to identify current issues faced by both detained immigrant children and pediatric clinicians, and suggest approaches to addressing these issues.

**Keywords:** child health; health disparities; immigration; children of immigrant families; detention

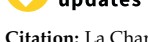



## 1. Introduction

The health and rights of immigrant children during and following detention are of paramount concern, particularly in light of recent United States (U.S.) governmental actions including family separations, prolonged federal detention, and legislative decisions stranding asylum-seeking families in Mexico (American Immigration Council 2021; Cheatham and Roy 2021; Todres and Fink 2020; Moore 2019). News reports describe episodes of pediatric communicable disease transmission and detainee deaths while in custody (Bhatt and Peeler 2020). Pediatric immigration detention hospitalizations are also of higher acuity and require longer lengths of stay compared to the average pediatric hospitalization (Nwadiuko et al. 2022). Case studies highlight the trauma and poor health conditions that children migrating to the U.S. experience during their migration and detention (Linton et al. 2018). Diverse research studies of former detainees (including immigrant adults detained in the US and pediatric and adult populations in other countries) substantiate the high burden of adverse and traumatic experiences as well as complex medical, educational, and psychosocial needs (Hanes et al. 2019; Dudley et al. 2012). As a result, pediatric clinicians and national organizations, including the American Academy of Pediatrics (AAP), continue to advocate for the rights and evidence-based care of immigrant children by publishing policy statements, advocacy reports, and clinical guidelines (American Medical Association 2020; Oberg et al. 2020; Linton et al. 2017). For example, these efforts call for limited exposure

to current Department of Homeland Security facilities, reduction of the custody period, establishment of comprehensive medical homes, and child access to free legal counsel (Linton et al. 2017).

Unfortunately, many of these policy recommendations have not been implemented, and children continue to face challenges in and following detention. Importantly, there is no known recent national survey of U.S. pediatric clinicians to understand their experience caring for immigrant children who were previously in government custody. Trusted relationships between the clinician, child, and family create space for children and families to share challenges navigating a new medical and legal system following custody release (Hoskins et al. 2021; Katzman and Christiano 2022). Pediatricians can draw attention to areas where new supports and collaborations are needed to prevent care and service delays and protect the rights of children. Survey findings may support advocacy efforts and policy recommendations by the AAP and others and highlight if and how prior policy recommendations and clinical guidelines are influencing the pediatric care of immigrant children. Overall, these findings could support ongoing efforts to protect the rights of detained immigrant children. Under Article 24 from the Convention on the Rights of the Child, children have the right to enjoy the "highest attainable standard of health" and access to health care services (UN General Assembly 1989).

Our research question was to explore whether current post-custody release processes, available clinical guidelines and resources were sufficient for pediatric clinicians to care for immigrant children previously in government custody. Specifically, our aims were to understand (i) the common health problems pediatric clinicians encountered, (ii) whether pediatric clinicians noted care delays, and (iii) clinician use of current clinical guidelines and resources. We conducted an online national survey and thematic analysis to describe clinicians' experience between 2014 to 2019 caring for immigrant children previously under government custody of U.S. Immigration Customs Enforcement (ICE), Customs Border Protection (CBP), and/or the Office of Refugee and Resettlement (ORR).

## 2. Compiling Pediatric Clinician Experiences

To understand the experiences of pediatric clinicians, we distributed a 66-question survey created in Qualtrics (Provo, UT, USA) focused on clinician demographics, delays in care experienced by immigrant children, health problems encountered and resources requested by clinicians. There was a mix of multiple choice and free response items. After validation by content experts and survey methodologists followed by pilot testing, the survey was distributed across 50 states and the District of Columbia. Participants were identified through their membership in select state chapters of the American Academy of Pediatrics, or in special interest groups for immigrant and refugee healthcare. It was distributed by email with reminders at the discretion of those professional organizations and interest groups. Since there was a low response rate and small sample size, we elected to perform a thematic analysis of the survey results. We reviewed and discussed the survey findings to come to a consensus regarding the themes that emerged from the results. This research project was acknowledged by the Johns Hopkins Institutional Review Board.

Of the 6200 clinicians who were sent the survey, we received 92 responses. Of the 92 responses, 82 respondents cared for immigrant children clinically, and of these, 34 cared for immigrant children previously or currently in government custody (Table 1). Most of the clinicians were attending physicians, but we also received responses from medical trainees, mid-level providers, and nurses. A majority of clinicians engaged in pediatric clinical practice and several others also had roles in research, education, administration, and community organizations. Most practiced community outpatient general pediatrics, but there was a wide range of practice settings and several pediatric specialties represented.

**Table 1.** Demographic Characteristics for Pediatric Clinician Respondents of National Survey, N = 82.

| **Healthcare Role** | |
| --- | --- |
| Attending | 67 (82%) |
| Resident, Fellow | 11 (13%) |
| Nurse, Physician Assistant, Retired, Other | 4 (4%) |
| **Specialty** | |
| Pediatrics or Internal Medicine and Pediatrics | 78 (95%) |
| Family Medicine or Psychiatry | 4 (5%) |
| **Practice Settings *** | |
| Community general outpatient | 31 (41%) |
| Academic general outpatient | 24 (32%) |
| Subspecialty | 18 (24%) |
| Academic general inpatient | 13 (17%) |
| Emergency Department | 7 (9%) |
| Community general inpatient | 4 (5%) |
| Urgent care | 4 (5%) |
| **States by Region** | |
| **West** | |
| Colorado | 34 (41%) |
| California | 2 (3%) |
| Oregon | 2 (3%) |
| **Midwest** | |
| Illinois | 16 (20%) |
| **Northeast** | |
| Maryland | 17 (21%) |
| Massachusetts | 2 (3%) |
| Rhode Island | 1 (1%) |
| **Southeast** | |
| North and South Carolina | 2 (2%) |
| **Southwest** | |
| Texas | 3 (4%) |
| No Response | 3 (4%) |
| **Care of Immigrant Child Held Currently or Previously in ICE/CBP/ORR Custody** | (N = 76) [a] |
| Yes | 34 (45%) [b] |
| No | 27 (35%) |
| Unsure | 13 (17%) |
| No Response | 2 (3%) |

[a] Only 76 were eligible for this question based on the skip pattern. [b] Only these 34 respondents were eligible for the remainder of the study. * Responses were not mutually exclusive. Due to rounding and non-mutually exclusive questions, the percentages for responses to a given question may not sum to 100%.

## 3. Immigrant Child Detention Is Likely Underrecognized by Pediatric Clinicians

Clinicians were not consistently identifying immigrant children who were previously held in ICE, CBP, and/or ORR custody. Some clinicians were unsure of whether they had cared for at least one child in their practice who had been held in ICE, CBP, and/or ORR custody for any period of time. One clinician described how "I have taken care of only a few patients that I knew had been detained but worry there were more and we did not have a systematic way of asking or thinking about it." Another respondent corroborated that clinicians were likely missing these children in their practice; "since returning to California, I have started to notice patients that I hadn't previously identified who have experienced ICE/CBP centers. They often presented to our urgent care for positive PPD* tests per report, but no details and refer vaguely to the test being done at a center in Texas." Similarly, some clinicians were not aware of the legal status of immigrant children in their practice.

For the clinicians who did learn that their patient had been in ICE, CBP, and/or ORR custody, there were a variety of sources used to acquire information regarding a

child's detention history. Most often, it was the result of the patient or family offering the information without prompting, or the clinician personally asking the patient or family. It was less likely that clinicians used a standard questionnaire or were contacted directly by a governmental agency. Even when a clinician knew that their pediatric patient was previously detained, a portion of them were unsure who accompanied them across the U.S. southern border or whether they were separated from an adult caretaker.

*Note: "PPD test" is a tuberculosis skin test.

### 4. Many Pediatric Clinicians Felt Ill-Equipped to Care for Immigrant Children Who Were Recently Detained

The clinicians saw a wide range of medical, psychological, and social conditions among their previously detained pediatric patients; most commonly they reported child exposure to adverse experiences, lack of routine health care, and mental health concerns (Table 2). The clinicians found that their pediatric patients were exposed to many traumatic experiences well beyond the conventionally described adverse childhood experiences (ACEs) including exposure to violence (most commonly), sex and labor trafficking, exposure to gangs and cartels, and separation from parents during their migration journey. Many infectious pathogens that are not endemic to the U.S. were identified in migrant children including *Helicobacter pylori*, *Strongyloides stercoralis*, *Schistosoma* species, *Ascaris lumbricoides*, *Vibrio cholerae*, *Entamoeba histolytica* and *Giardia*. Several clinicians also saw medically complex children including those with congenital heart disease, developmental delay, malnutrition, glomerulonephritis, epilepsy, and history of stroke. Layered onto the medical complexity were high social needs including housing and food insecurity, limited transportation, lack of school enrollment, and lack or underinsurance. Specific legal barriers were also identified including a lack of access to a lawyer and challenges navigating the legal system, particularly in applying for asylum or other legal protections.

In the context of this medical and social complexity, most clinicians did not feel they had sufficient resources to provide appropriate care for these immigrant children and families (Table 2). One clinician requested additional background information on governmental custody settings, medical care provided in these settings, and formalized training on evidence-based medical care of immigrant children. Although there are guidelines for the care of refugee children, including the CDC's Domestic Refugee Screening Guidelines (Centers for Disease Control and Prevention 2021) and the AAP's Toolkit for the Care of the Immigrant Child (Meneses et al. 2021), many clinicians were not familiar with these guidelines, nor were they using them in their clinical practice (Table 2). Additional supports in their clinical setting were also requested including legal, interpretation, social work, mental health, physical/occupational/speech therapy, and resources to address the social determinants of health (Table 2). The clinicians acknowledged that these children have unique needs requiring additional services. For instance, the "child and guardian were unaware and unsure how to apply for legal status and were scared to do so." Another challenge was obtaining medical records for their pediatric patients. As one respondent noted, there was "insufficient documentation to be able to determine if any of [standard healthcare screening was done] so most things need to be done but unclear if they already had been done and typically the family is not sure either." Most clinicians were unsure if their patients had a standardized medical evaluation performed during custody and if so, how to obtain medical records.

Lastly, concerns emerged regarding delays in connecting children to needed primary medical care while in and following detention. Missed diagnoses while a child was in detention included infectious and non-infectious conditions, gynecologic-related problems, and mental health conditions. Clinicians reported a case of "post-traumatic stress disorder that was misdiagnosed as bipolar disorder and mistreated", "headache due to hypertension that was thought to be a [just a mild] headache", and "unrecognized acute kidney injury that lead to hypertension that then lead to seizure." Studies support that children previously detained had higher acuity and length of stay when admitted to the hospital compared to

national averages (Nwadiuko et al. 2022). Once released from custody, clinicians witnessed delays in establishing care with a primary care clinician due to cost, lack of transportation, fear of deportation, language barriers, lack of understanding of where to access pediatric health services, and fears related to the public charge policy.

**Table 2.** Clinician experience caring for children who have been in the U.S. Immigration Customs Enforcement (ICE), Customs Border Protection (CBP) and/or the Office of Refugee and Resettlement (ORR), N = 34.

| **Further Resources Needed to Care for Children Previously in Custody?** | |
| --- | --- |
| Yes | 16 (47%) |
| No | 9 (26%) |
| No Response | 9 (26%) |
| **Resources Needed for Care of Immigrant Children *** | (N = 16) [a] |
| Legal assistance | 15 (94%) |
| Mental health support | 15 (94%) |
| Financial supports for children and families | 14 (88%) |
| Health insurance coverage | 14 (88%) |
| Medical records from ICE, CBP and/or ORR | 14 (88%) |
| Housing supports | 13 (81%) |
| Social work supports | 12 (75%) |
| Education about detention for clinicians | 10 (63%) |
| Primary care access | 10 (63%) |
| Education about healthcare in detention for clinicians | 9 (56%) |
| PT, OT, SLP | 9 (56%) |
| Need for an interpreter | 7 (44%) |
| Education on immigrant health issues for clinicians | 4 (25%) |
| **Health Issues Encountered *** | |
| Trauma/Adverse childhood experience exposure | 21 (62%) |
| Lack of routine health maintenance | 19 (56%) |
| Mental health problems | 17 (50%) |
| Dental problems | 16 (47%) |
| Legal problems | 16 (47%) |
| Socioeconomic problems | 16 (47%) |
| Developmental problems | 11 (32%) |
| Gastrointestinal problems | 10 (29%) |
| Infectious disease problems | 10 (29%) |
| Nutritional problems | 6 (18%) |
| Neurologic problems | 5 (15%) |
| Cardiovascular, Hematologic, Endocrine problems | 5 (15%) |
| Obstetric/gynecologic problems | 4 (15%) |
| Musculoskeletal problems | 4 (12%) |
| Environmental exposure (e.g., lead), Renal problems, Other | 3 (9%) |
| No Response | 6 (18%) |
| **Aware of CDC Domestic Refugee Screening Guidelines** | (N = 64) [b] |
| Yes | 27 (42%) |
| No | 34 (53%) |
| Unsure | 3 (5%) |

[a] Only the 16 who answered *yes* to needing resources were eligible for this question. [b] 64 respondents were eligible for this question about CDC guidelines. Respondents who saw immigrant children, regardless of whether they were previously detained, were eligible for this question. * Responses were non-mutually exclusive. Due to rounding or non-mutually exclusive questions, the percentages for responses to a given question may not sum to 100%.

## 5. Reflections and Recommendations

Although the AAP, the United Nations, and many advocacy organizations have worked to reform or end child detention, it is still taking place today in the U.S. If current trends persist, as we believe they will, there will continue to be large numbers of children

and families who attempt to cross the U.S.-Mexico border; these children and families will continue to overwhelm current infrastructure to expeditiously process children to be released to a safe space that is appropriate for children and that addresses their healthcare and other essential needs using an evidence-based approach. We must work to create systems to support children throughout their migration journey that prioritize their health, well-being and safety and keep their best interests in mind.

After eliciting comments from pediatric clinicians, we found new evidence that clinicians may be failing to recognize immigrant children who were previously detained. If clinicians are not aware that their pediatric patients were previously in government custody, then they are unable to identify and address their specific healthcare, social, and legal needs. Mounting evidence demonstrates that even short periods of time in detention are related to higher rates of psychiatric conditions in individuals of all ages (Dudley et al. 2012). Therefore, there may be missed opportunities to ensure that children are getting the mental health support they need. Appropriate identification of formerly detained immigrant children would also allow health professionals to document histories, physical and mental health exam findings that can be used to bolster asylum claims (Dudley et al. 2012). Investigations are needed to identify culturally sensitive and supportive approaches to screen immigrant children in healthcare settings for a history of detention or recent migration that are not retraumatizing, do not contribute to mistrust, and do not put them at risk for negative consequences due to their legal status. As screening children in healthcare settings for adverse childhood experiences (e.g., child maltreatment, household dysfunction) and social determinants of health becomes standardized, this may offer an opportunity to embed questions related to migration and detention. Critical structural changes are also needed that allow for increased communication and transparency between governmental agencies and medical providers. For example, we recommend the development of a standardized approach by ORR to provide copies of medical records, including vaccine records and laboratory testing results to each child when released from custody and a list of basic instructions for sponsors or legal guardians in their primary language to follow to link the child to a medical home, legal services, and school enrollment. Also, ORR should develop a standard medical sign-out sheet with instructions for the sponsor/guardian to provide to the accepting medical provider. Information should include links to ORR's site on how to obtain medical records, medical testing that was completed while in ORR custody, and recommendations to follow CDC Domestic Refugee Screening Guidelines to complete appropriate medical screening (Centers for Disease Control and Prevention 2021).

Our findings also highlight that clinicians need additional training in standardized, evidence-based medical screening of immigrant children as well as legal relief available for migrant children, and how to link families to legal services. There are models for medical-legal partnerships that embed lawyers as specialists in healthcare settings, but these are not currently widespread (Mandelbaum 2020). We call for further cross-sector collaboration between medical and legal professionals and institutions, along with funding mechanisms to facilitate the development and expansion of such partnerships. From these collaborations, existing curricula could be adapted and disseminated to standardly teach medical social workers, nurses, medical trainees and clinicians about legal issues specific to immigrant populations, and how they can support and direct their patients to legal resources. Pediatric clinician survey respondents also advocated for the development and expansion of medical homes with mental health, legal, and social work supports to care for children who were previously detained to optimally care for their specific needs. These primary care settings with expanded services reinforce the AAP's call for comprehensive medical homes (Linton et al. 2017).

Despite the presence of existing clinical guidelines to address the unique health needs of immigrant children, there remains a need for further knowledge sharing with pediatric clinicians (Centers for Disease Control and Prevention 2021; Meneses et al. 2021). This extends to pediatric clinicians and staff not only in medical homes, but also in urgent care clinics and emergency departments, since immigrant children may first present to

these non-primary care settings (Linton et al. 2017). Of note, some healthcare settings have support tools with immigrant screening guidelines embedded into electronic medical records for clinicians to access readily when needed (Orenstein et al. 2017).

Finally, we call for rigorous studies with larger nationally representative samples to systematically evaluate formerly detained children to determine whether they are safe, receiving standard healthcare, and ensure that their collective social determinants of health, including legal representation, are being met. This data can inform systems of care improvements and policies to support immigrant children and their families.

## 6. Conclusions

Immigrant children who experience detention have specific healthcare needs, many of which clinicians do not feel equipped to properly address. By sharing the experiences of pediatric clinicians, we hope to spur conversations across sectors for combined advocacy for the rights of immigrant children to receive special protections and investment in medical homes that provide evidence-based medical care and address all their needs. This can be done as we work toward the end child detention for children accompanied by a legal guardian and the swift processing of unaccompanied children in favor of safe and vetted placements within the community where appropriate sponsors or legal guardians are residing and being evaluated.

**Author Contributions:** Conceptualization, J.L.C., E.W.T., J.R., J.Y. and K.H.; methodology, J.L.C., E.W.T., J.R., J.Y. and K.H.; software use, J.L.C.; validation, J.R. and J.Y.; data curation, J.L.C. and N.G.; formal analysis; J.L.C.; data interpretation; J.L.C., E.W.T., J.Y. and K.H.; writing—original draft preparation, J.L.C.; writing—review and editing, J.L.C., E.W.T., J.R., J.Y., N.G. and K.H.; visualization, J.L.C. and N.G.; supervision, K.H.; project administration, J.L.C. All authors have read and agreed to the published version of the manuscript.

**Funding:** This research received no external funding. E.W.T.'s time was supported by the NIH NIAID K08AI139371. J.L.C.'s time during the manuscript revision phase was supported by the National Clinician Scholar Program at University of California, Los Angeles, and Cedars-Sinai Medical Center.

**Institutional Review Board Statement:** The study was conducted in accordance with the Declaration of Helsinki, and approved by the Institutional Review Board (or Ethics Committee) of Johns Hopkins (protocol code IRB00221082 and date of approval).

**Informed Consent Statement:** Participants were informed that they provided consent through their voluntary participation in the survey.

**Data Availability Statement:** The data presented in this study are available on request from the corresponding author. The data are not publicly available due to privacy and ethical restrictions. There is the potential for identification of the respondent given the small sample size and respondent commentaries.

**Conflicts of Interest:** The authors declare no conflict of interest.

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
