# Peer review of "Understanding the Healthcare Needs of Immigrant Children Currently and Previously in Government Custody: A Narrative"

_laws, 2017_

Round 1

Reviewer 1 Report

This is interesting and dynamic work. Despite having researched in this area for some years, I have not seen this particular issue presented in this way. The difficulties that pediatricians are seeing as front-line practitioners for child migrants is critical research for advocates to understand. This is poignant and time sensitive. My biggest suggestion is that because the sample size is so low due to low response rate, can anything be done to supplement the sample size and update the piece? Further, did the research instrument capture what pediatrians felt or experienced in this work? I would welcome that front-line perspective.

Line 98 needs an edit on "caretaker."

Reviewer 2 Report

The article investigates paediatricians experiences with children in migration and identifies paediatricians as an important entry point for strengthening the rights of children in migration (my interpretation). Overall, it  is important research and might be strengthened by more rigorous description of its contribution to literature, design and analysis of the findings and recommendations.

Acknowledge children's rights

Though in a special issue on 'protecting children's rights' it however does not mention children's rights. Though the US is not a party to the Convention on the Rights of the Child, under article 24 of the International Covenant on Civil and Political Rights it has a legal obligation to ensure children's right to special measures of protection. The Human Rights Committee has interpreted this to include social and economic measures including presumably healthcare.

In most cases, however, the measures to be adopted are not specified in the Covenant and it is for each State to determine them in the light of the protection needs of children in its territory and within its jurisdiction. The Committee notes in this regard that such measures, although intended primarily to ensure that children fully enjoy the other rights enunciated in the Covenant, may also be economic, social and cultural.

 Human Rights Committee ‘Rights of the Child (Art 24)’ General Comment No 17 4 July 1989 UN Doc CCPR n/a  para 3   

Though this may be beyond the expertise of the authors, the article's raison d'être and conclusions might still be tied to and loosely framed in terms of children's standing as rights-holders. 

Review and evidence assumptions 

Where is the evidence that paediatricians are trusted professionals? I suggest removing the reference to 'trusted'. Without data, it is highly subjective and overlooks the power differentials between clinicians and those in their care. 

Review language from a rights perspective

Review language and avoid language such as 'influxes', the children etc. Perhaps refer to children and young persons under 18, if that works.

Strengthen analysis of current literature

The article might be strengthened by reducing the very long (and widely known) introductory description about the situational context — and developing the introductory analysis of current literature. Where is the originality? the contribution to scholarship? What are the focuses of the current literature in this area? Where does this paper fit in?  The Academy's recommendations might be referred to here (see below)? 

Clarify research objectives and questions 

Though the analysis of the methodology is fairly comprehensive, it might be improved by clearly stating and clarifying first the research questions and objectives.  

Clarify process of data collection 

How was the data disaggregated? what were the demographics of clinicians and children? how old were the children? what gender? etc

Where are the tables? research results? 

Though tables 1 and 2 are referenced I could not find them? Also  I did not understand the distinction between 'cared for children clinically and 34 cared for children previously or currently in government custody '. 

Review, distinguish and clarify conclusions 

What are the key findings and recommendations of this research? How are they different from the Academy's? What are the distinguishing findings and recommendations of this research? development of tools to identify children who have been detained? child rights training? legal training? collaboration between medical and legal practitioners?    

Perhaps the reference to recommendations of American Academy of Pediatrics should be included in the introductory review of current literature. The conclusion then may refer back to  Academy's recommendations, noting briefly the Academy's complement  the findings of this research. What is more important here is your/ this research's findings and recommendations.
